# The compressive strength of crumpled matter

Andrew B. Croll [1,2], Timothy Twohig [1] & Theresa Elder[2]

Crumpling a sheet creates a unique, stiff and lightweight structure. Use of crumples in engineering design is limited because there are not simple, physically motivated structure-property relations available for crumpled materials; one cannot trust a crumple. On the contrary, we demonstrate that an empirical model reliably predicts the reaction of a crumpled sheet to a compressive force. Experiments show that the prediction is quantitative over 50 orders of magnitude in force, for purely elastic and highly plastic polymer films. Our data does not match recent theoretical predictions based on the dominance of building-block structures (bends, folds, d-cones, and ridges). However, by directly measuring substructures, we show clearly that the bending in the stretching ridge is responsible for the strength of both elastic and plastic crumples. Our simple, predictive model may open the door to the engineering use of a vast range of materials in this state of crumpled matter.

[1] Department of Physics, North Dakota State University, Fargo, ND 58102-6050, USA. [2] Materials and Nanotechnology Program, North Dakota State University, Fargo, ND 58102-6050, USA. Correspondence and requests for materials should be addressed to A.B.C. (email: andrew.croll@ndsu.edu)

I t is not uncommon to take out the stress caused by an unpleasant letter by crushing the paper into a tight ball. Such action, while perhaps stress relieving, causes a problem for the eventual trash collection because the letter now takes up more volume than it needs to and is not easily compacted further (after hand crushing, a paper ball is estimated to be 75% air)[1]. From an alternate point of view, the low-density crumpled letter is now a potentially useful material with properties comparable to those of a solid foam or engineered lattice[2–5]. What is truly remarkable is the ease with which the crumpled material is produced. From films of materials, such as metals, natural or synthetic polymers and graphene, to systems as complex as thin electronic circuits can all be readily produced and crumpled[1,6–11]. Unfortunately, crumpled matter is rarely used because it is still so poorly understood in comparison with engineered structures and foams. For example, it remains unclear how much load, $F$, can be supported by a square sheet of dimensions $L \times L \times h$ crumpled into an approximately spherical ball of radius $R$.

Despite the complexity, crumpled sheets (as well as origami structures) are comprised of only a handful of building block structures. The four most dominate, the bend, the fold, the d-cone, and the stretching ridge are shown below in Fig. 1 below. Slender systems are easily bent due to the tiny energetic cost of bending when compared to stretching. The ratio of energies is reflected in the high Föppl-von Kármán numbers typical of sheets ($\gamma = L^2/h^2$, where $h$ is the film thickness). If the radius of curvature of a bend is decreased to the scale of the sheet thickness, the result is called a fold[12–14]. Folds are primarily a result of plasticity, meaning energy loss has occurred and memory has been created in the sheet (typically required for origami). Confining a sheet in three dimensions (as in crumpling a ball) leads to constraints on the sheet that can no longer be satisfied by bending alone. Confinement forces a sheet to stretch. Stretching comes at a high cost, so the system minimizes the total amount of stretching present. In thin, marginally confined systems, stretching is typically localized to a (near) singular point known as a developable cone (d-cone)[15–18]. Finally, if confinement is increased and two d-cones are created in a sheet, they are linked by a structure known as a stretching ridge[19]. In this case, the sheet cannot join the two d-cones and remain developable; it must also stretch along the ridge. Intuitively, theoretical models of crumpling are often developed from these simple structural building blocks; however, such models have not yet delivered quantitative prediction (See Supplemental Discussion)[1,8].

In this manuscript, we discuss experiments designed to unambiguously determine the force response of crumpled materials and to clarify which underlying structures dominate. The experiments use thin films created with a range of thicknesses (from 100 nm to 1 mm) from two very different but well-characterized polymeric materials, namely, polycarbonate (PC), a glassy polymer with a modulus of 1.6 GPa and polydimethylsiloxane (PDMS), an elastomer with a modulus of 1.69 MPa[20]. To simplify the results presented here, the PDMS is additionally treated to reduce its adhesion. Additional sample preparation detail can be found in the Methods section. The result is access to a vast range of $\gamma$s ($10^4$–$10^{12}$) and, more importantly, to vastly different material response (plastic or elastic). The films are crumpled by hand and loaded between two parallel glass plates, which comprise the compression cell. The film is observed with laser scanning confocal microscopy, leading to direct observation of the internal crumple structure during the entire test cycle. Single ridges are also tested in compression and observed with confocal microscopy. Remarkably, we find that the compressive forces on both crumples and ridges are well described by a similar power law function. We conclude that ridges are minimal crumples and serve only to divert energy into sharper bending as they are compressed.

## Results

**Crumpled polymer films.** Typical data are shown in Fig. 2 below, where the force response of the two materials differs as one might expect. The elastic PDMS data show a smooth indentation and retraction curve, with little hysteresis. On the other hand, the PC film shows significant hysteresis. Such hysteresis is often attributed to plastic losses, which occur during compression. The contrast between the two materials is also clear post experiment. The PDMS films recover their flat initial state and the PC films remain crumpled (as does paper).

When initially crumpled, films take on an approximately spherical shape. However, when confined by the parallel plates of the cell, the crumpled sheet assumes the shape of a rough cylinder of radius $R$ and height $H$ (Fig. 2b.). An effective modulus can be determined from the force data and the cylindrical geometry, and an effective engineering strain can be determined from the plate displacement. The data can then be meaningfully summarized in Fig. 2f. Individual experiments show very different trajectories, but the average forms a trend that is continuous through both plastic and elastic crumples. To guide the eye, curves for ideally stretching structures (red) and ideally bending structures (black) are shown. The latter forms a limiting value for crumpled matter regardless of specific material properties, strongly suggesting that bending is important regardless of $\gamma$.

To understand the origin of the behaviour shown in Fig. 2f, we return to the force measurement itself, which is free of any geometric assumptions (Fig. 2e). Force measurements for both materials are well fit by a power law of the form $F = F_0 H^{-\alpha}$, in agreement with many other experiments and simulations[1,6–10,21–24]. With PDMS, the exponent $\alpha$ converges to an average of 2.8 though the error is not insignificant (the standard deviation is $\Delta\alpha = 0.5$). PC, though well fit with a power law, yields a wide range of exponents ($\alpha = 7.7 \pm 5$ upon compression and $\alpha = 14.0 \pm 13.6$ during retraction). Clearly, $\alpha$ is not well defined by a single value and needs to be considered statistically (see Fig. 3). We note that the PDMS exponents appear to follow a normal distribution, whereas the PC exponents appear similar to a log-normal distribution. It is likely the exact value of the exponent is related to the underlying statistical distribution of rigid elements in the structure[11,25–27].

The resistance of the crumple to compression was initially suggested to be directly related to the forced stretching occurring

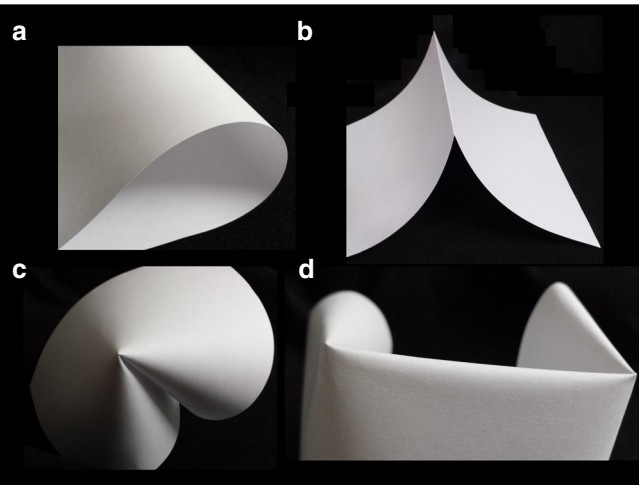

**Fig. 1** Four fundamental building blocks of origami and crumpled matter. **a** A bend. **b** A fold. **c** A developable cone. **d** A stretching ridge formed between two developable cones

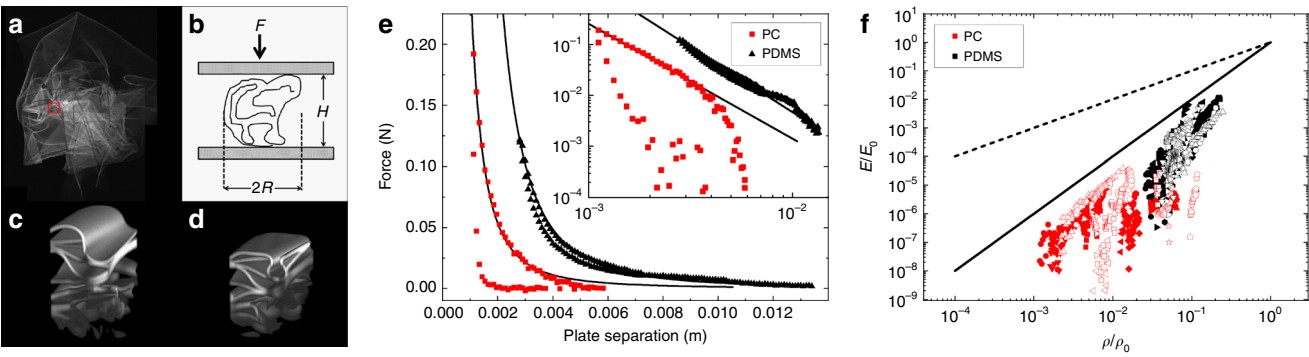

**Fig. 2** Compression measurements of crumpled matter. **a** Confocal microscopic image of a crumpled (69.5 mm × 46.1 mm × 79.4 μm) polydimethylsiloxane (PDMS) sheet in index-matching glycerine/water mixture. **b** Basic geometry of crumple compression experiment. **c** A three-dimensional rendering of the confocal data from the 1.3 mm × 1.3 mm red square indicated in **a**, in this case $H = 2.8$ mm. **d** Same location as c. under a compression of $H = 1.8$ mm. **e** Force displacement data for a 43 mm × 40 mm × 86.5 μm PDMS film (black triangles) and a 16 mm × 22 mm × 2 μm thick polycarbonate film. Inset shows the same data on a log/log axis. Both data sets are well fit by the power law $F = F_0 H^\alpha$, where $F_0$ and $\alpha$ are fit constants. **f** Normalized modulus data from several different crumpled polycarbonate (PC) (red—various symbols) and PDMS (black—various symbols) films plotted as a function of normalized density. Specifically, $E = (F/\pi R^2)/[(H - H_0)/H]$ and $\rho = L^2 h/H\pi R^2$, where $L$ is the average length and width of the crumpled sheet, and $H_0$ is the cell gap at the beginning of compression. The normalization constants $E_0$ and $\rho_0$ refer to the Young's modulus and density of the material making up the film (bulk PC or PDMS), respectively. The two solid lines are the ideal stretching (dashed) and ideal bending (solid) limits[2]

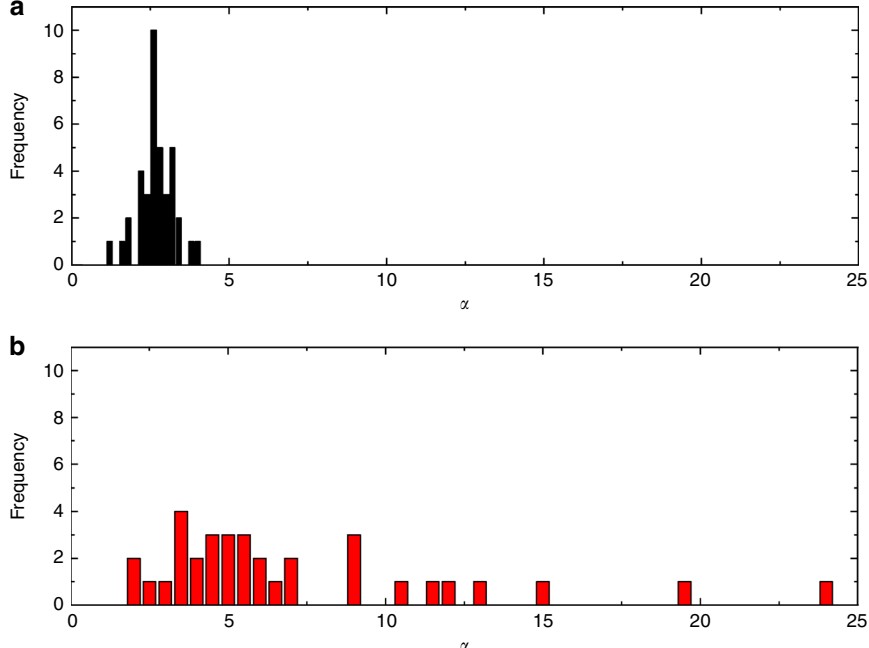

**Fig. 3** Histograms of measured power law exponents. **a** Polydimethylsiloxane (PDMS) exponents (black) and **b** polycarbonate (PC) exponents (red). PDMS data approximate a normal distribution, whereas PC data does not. $N = 38$ for PDMS and $N = 34$ for PC

in the many ridges within the crumple[1,19]. Using energy scaling arguments developed for the ridge and neglecting any other interactions (including self-avoidance of the sheet), it was predicted that $F_0 \sim E h^{8/3} L^{16/3} R^{-10/3}$ and $\alpha \sim -8/3$ where $E$ is the film's Young's modulus, $h$ is the film thickness, and $L$ and $R$ are defined above. Simulations were developed to test this hypothesis with mixed results. Vliegenthart and Gompper using a mesh of spring-linked nodes and a dimensional argument found an exponent of ~14/9 with phantom sheets (matching the ridge model) but a value of ~2 with more realistic self-avoiding sheets[22]. It is interesting that the exponents measured for the simulated self-avoiding sheet, when input into the dimensional scaling, implies that a single effective bend is dominant (as a single bend scales as $F \sim E h^3 L/H^2$)[20].

The experimental measurements reported in this letter do not agree with the ridge model predictions. The exponents measured for PDMS films do closely match the ridge-model exponent of 8/3; however, comparing the amplitude ($F_0$) yields only weak correlation and several orders of magnitude error in scale (see Supplementary Discussion). The disagreement in the PDMS data may be due to the lower Föppl-von Kármán numbers accessed by the experiments ($\gamma \sim 10^4 - 10^7$), as the asymptotic scaling on which the model is based is only valid above $\gamma = 10^8$. The PC films are well within the asymptotic limit ($\gamma_{PC} \sim 10^5 - 10^{12}$) but give exponents that are far too large. Experimental $F_0$ values are also off by several orders of magnitude with PC crumples.

More recently, Deboeuf et al. suggested an alternative model based on energy storage in the irrecoverable plasticity occurring

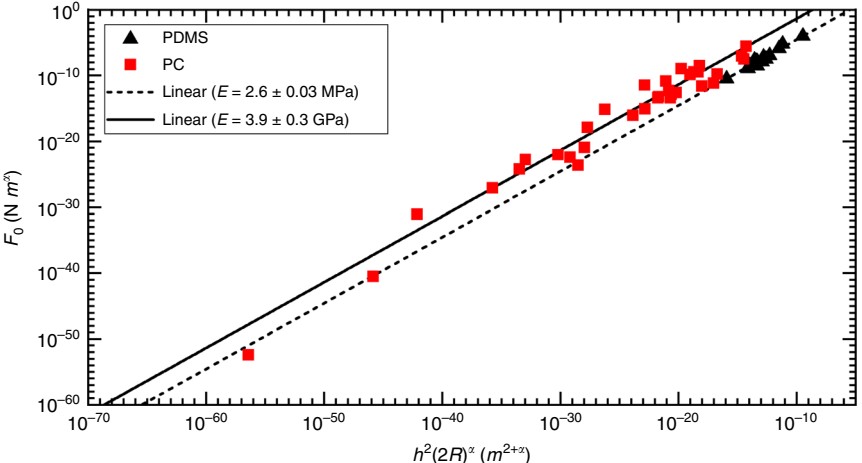

**Fig. 4** Force amplitude plotted against $h^2(2R)^\alpha$. All data are well fit by Eq. 1, yielding a measured value for Young's modulus for each polymer that agrees with traditional mechanical measurements

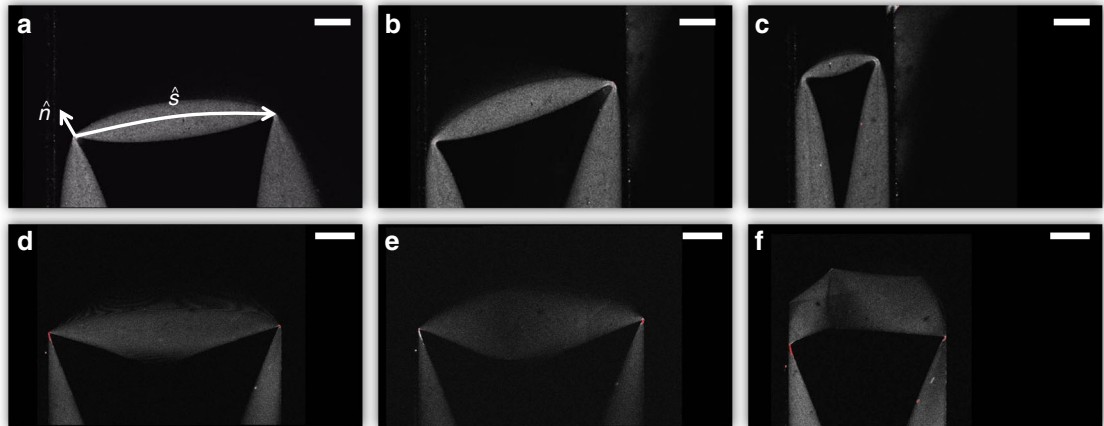

**Fig. 5** Confocal microscopy of the compression of a stretching ridge. **a** Geometry of a 58.5-μm-thick polydimethylsiloxane ridge confined to a $X = 11.3$ mm gap. Scale bar represents 2 mm. **b**, **c** show the effects of additional compression where $X = 6.3$ mm and $X = 2.8$ mm, respectively. **d–f** show similar compression of a 9.3-μm polycarbonate film at $X = 11.7$ mm, $X = 11.0$ mm and $X = 7.7$ mm, respectively. In this case, the ridge buckles, forming two new d-cones as it is compressed. Again the scale bar represents 2 mm

in sharp folds[8]. The model allows several different values for $\alpha$, ranging from 1 to 4 depending on the underlying structure and type of compression. Additionally, the model predicts an amplitude of $F_0 = Eh^2L^\alpha$. The fold model was validated experimentally through the crushing of cylindrically bent (rolled-up) sheets and sheets confined in three dimension by a wire mesh. Exponents were found to depend on geometry and material properties[8,10]. The fold model applied to our data shows qualitative agreement, but once again, a quantitative error of several orders of magnitude is revealed (see Supplementary Discussion). Furthermore, the exponents measured for PC are beyond what is allowed by this model and confocal observation of PDMS shows very few sharp folds.

Interestingly, a similar relationship,

$$F = Eh^2 \left(\frac{2R_0}{H}\right)^\alpha,\qquad(1)$$

provides an excellent fit to the data. To be clear, Eq. 1 is an empirical model that is not necessarily related to folds. We will discuss possible origins of this scaling below. The model is best demonstrated in Fig. 4 where $F_0$ is plotted against $h^2(2R)^\alpha$, and the slope of a straight line is the material's Young's modulus. The plot shows that both PDMS and PC data fall along separate linear

trends, the slope of each is $2.6 \pm 0.03$ MPa and $3.9 \pm 0.3$ GPa, respectively. Given the independently measure modulus of these materials is $1.7 \pm 0.05$ MPa and $1.6 \pm 0.1$ GPa, and the linearity spans 50 orders of magnitude, Eq. 1 seems reasonably reliable.

It is important to note that no attempt to create a repeatable crumple was made in the experiment, suggesting that the wide range of exponents observed is simply related to the wide range of configurations explored. The exponent must then be related to the network structure of rigid elements, not all of which are initially load-bearing. What the rigid elements are is not yet clear, because both ridge- and fold-based scaling models arrive at different relationships than Eq. 1.

**Compression of isolated ridges.** In order to clarify what (if any) underlying structure is dominant, we have conducted compression experiments on isolated ridges. The experiment is once again conducted under the confocal microscope, and once again complete structural information is recorded during the experiment. Figure 5 shows typical experimental images collected during compression of PDMS (Fig. 5a–c) and PC (Fig. 5d–f) thin films configured in the ridge geometry. During compression a PDMS ridge smoothly shrinks in length, whereas the PC ridge does not observably change until it catastrophically buckles (Fig. 5e–f). In

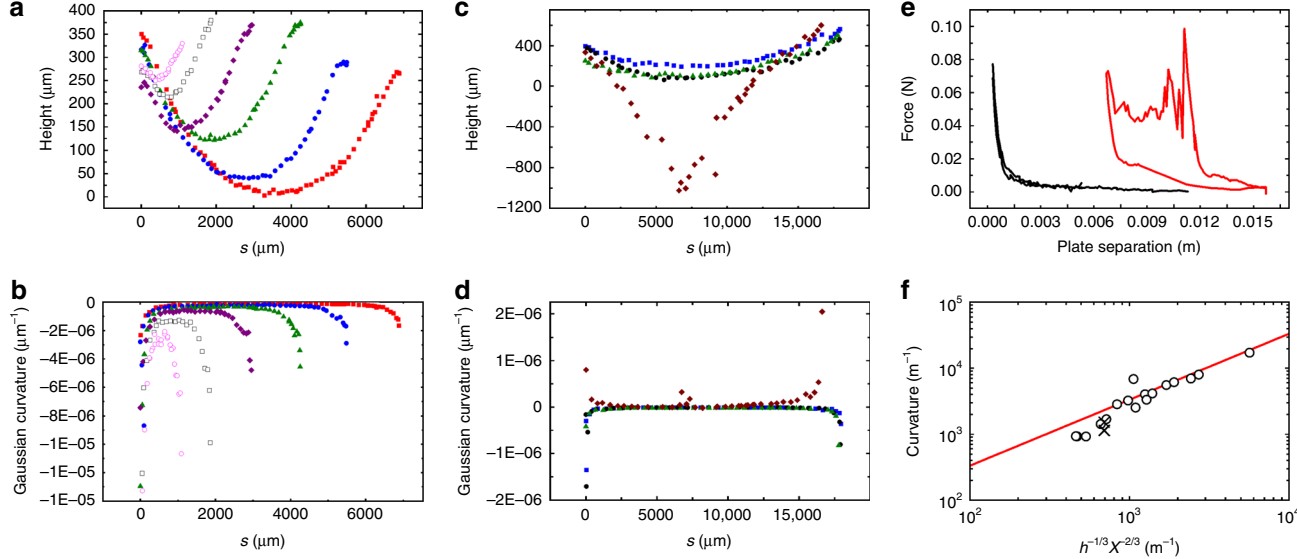

**Fig. 6** Geometry of a stretching ridge. **a** Ridge height of the polydimethylsiloxane (PDMS) sample shown in Fig. 5a–c as a function of s, a coordinate running along the peak curvature of the ridge. All quantities are obtained by fitting a parabola to the film surface in a section cut orthogonal to s. The film surfaces are observed directly with confocal microscopy. **b** The Gaussian curvature of the ridge as a function of s. **c** Height data of the polycarbonate (PC) sample shown in Fig. 5d–f. **d** Gaussian curvature of the PC film as a function of s. Note that there is little change in curvature until the ridge buckles. **e** Force displacement data for the PDMS (red) and PC (black) samples. **f** Curvature at the ridge midpoint vs $h^{-1/3}X^{-2/3}$. The linear trend shows the agreement between the geometry assumed in ref. [19] and both PC and PDMS experiments

this work, we limit our discussion to data collected prior to the buckling of the ridge.

Arranging a coordinate, s, along the ridge's crest, the ridge height and curvature as a function of position can be determined (Fig. 6a–d). PDMS shows a smooth change in height and curvature during compression and PC shows little change before buckling. Force as a function of plate separation, $H$, is shown in Fig. 6e. PDMS shows a smooth monotonic trend and little hysteresis. PC shows an increasing monotonic trend similar to PDMS before buckling and a dissipative regime as two additional d-cones form during buckling[28].

The energy stored in a ridge is theorized to be primarily stored in the stretched material along the ridge peak[19]. This is evident in Fig. 6a, where the ridge crest drops below the horizontal, meaning its length is longer than is needed to directly connect the two d-cones. Using the strain determined geometrically, the total (stretching plus bending) energy can be minimized to yield the curvature, $\kappa = h^{1/3}X^{-2/3}$, and total energy $U_{\text{ridge}} \sim Eh^{8/3}X^{1/3}$[19]. Here $X$ is the total distance between the two d-cones (approximately the plate separation, $H$). The curvature prediction is directly verified by the measurement as we show in Fig. 6f, where we plot curvature (directly measured by confocal microscopy) against $h^{8/3}X^{-2/3}$. The linear trend shown in the figure is the prediction with no free parameters. The compressive force can be found by comparing the total energy with the work done compressing the structure, $F_{\text{ridge}} = Eh^{8/3}X^{-2/3}$. Despite the geometry being correct, the predicted force does not match measurements for either material.

Compression pushes the two d-cones together. In PDMS, this occurs smoothly and one d-cone is eventually annihilated. This means that the retraction curve is the mechanical response of a sheet with a single d-cone and no ridge. The low hysteresis shows that the measurement is not significantly probing the ridge; the bending in the rest of the structure is more important (reflecting the low Föppl-von Kármán numbers accessed by the PDMS experiments). On the other hand, in PC the d-cones are fixed at the location they are initially created; the d-cones cannot move

rendering the ridge's minimal energy irrelevant to the overall energy storage during compression. This means that, once again, the bending in the structure is all that changes until the ridge buckles. In both crumples and ridges, the experiment primarily probes bending near the d-cone cores.

The ridge force–displacement data are reminiscent of the crumples and is likewise well fit by a power-law relationship $\left(F_{\text{ridge}} = F_0 H^{-\alpha}\right)$. Again, variation is observed in the power law exponents themselves ($\alpha = 2.4 \pm 1.7$ for PDMS ridges and $\alpha = 9.3 \pm 6.1$ for pre-buckling PC ridges). Given the similarities, we propose that Eq. 1 can also be used to describe the ridge mechanics. Replacing the initial crumple size ($R_0$) with the initial ridge length ($X_0$), the relation can be expressed as:

$$F = Eh^2\left(\frac{X_0}{H}\right)^{\alpha}. \tag{2}$$

We demonstrate the validity of Eq. 2 in Fig. 7 where we plot $F_0$ against $h^2X_0^{\alpha}$. Both PC and PDMS data show linear trends and have slopes of $1.2 \pm 0.1$ GPa and $6.2 \pm 0.1$ MPa respectively. While the data only spans 40 orders of magnitude in $F_0$ in this case, the agreement is still quite sound. Apparently, the ridges serve as a minimal crumple.

## Discussion

That a single model can simultaneously account for the compression of a crumple (Eq. 1) and a ridge (Eq. 2) is strong evidence that the two structures are similar. The basic hypothesis that crumples can be considered to be a collection of ridges seems to be correct. However, the balance of stretching and bending in the ridge does not explain the observed scaling. We tentatively conclude that the ridges are only involved indirectly; they funnel energy into the available soft modes (bending), which progressively stiffen until the ridge structure collapses (either by annihilation or by buckling).

The curvature in the centre of the ridge is not the largest curvature in the structure, so it is not surprising that it is not the largest contribution to the compaction process (although it does

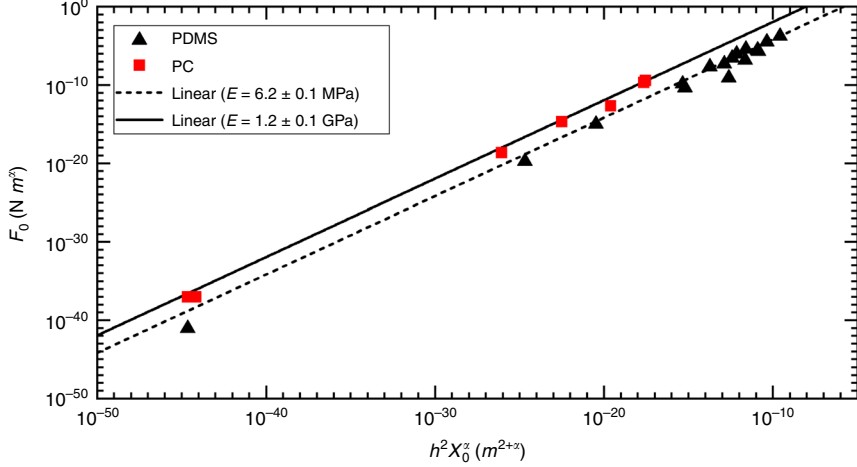

**Fig. 7** Force amplitude plotted against $h^2(X_0)^\alpha$. All data are well fit by Eq. 2, yielding a second measured value of Young's Modulus for each polymer. The agreement is fair, but error in modulus is underestimated by the fit statistics

still explain the local ridge shape, Fig. 6f). The leading $Eh^2$ term in the scaling of Eqs. 1 and 2 is another hint as to the underlying physics. As pointed out by Deboeuf et al., this is a natural energy scale for structures in which the radius of curvature is comparable to the sheet thickness[8]. We note that we observe no folds in the ridges of our experiment, so the premise that folds are important is misleading. The only place curvatures approach the sheet size is in the d-cone cores themselves.

The remaining detail necessary for understanding the observed scaling is an estimate of the extent of the high curvature region. Figure 6a, b (or Fig. 6c, d) show that the curvature drops smoothly from the high values in the d-cone core over a significant distance along the ridge, meaning that the d-cone cores are not as localized as might be expected from current theories. In fact, the high curvature appears to span the ridge, meaning the relevant length scale in the problem must be of the order of the length of the ridge itself. This means that a perturbation to the structure can be described in terms of a function, $f$, of the unitless number $X_0/H$, which is evidently a power law ($f(X_0/H) = (X_0/H)^\alpha$). A more comprehensive model of the d-cone motion is needed to theoretically predict the function $f$ from first principles.

The bending-dominated crumple hypothesis is further strengthened by the limiting behaviour observed in the normalized modulus vs normalized density data shown in Fig. 2. From this point of view, we can see that individual crumples stiffen with compression but only up to the limit set by bending. No experiments observed crumples reaching the stretching limit, which would be the case if the load was being directed primarily into stretching modes within the crumple. The structures never breach the limit set by bending; they are structures that deform primarily through increased bending.

Crumples are complex but are trustworthy materials. They are structures dominated by bending at many different length scales similar to a single stretching ridge. Importantly, they can be described quantitatively by a power law relating force to compression. The detailed prediction of the power law exponents remains to be determined, though statistically measured values are sufficient for predicting system behaviour. Not only can the load held by a crumple now be quantitatively predicted, but the model also allows a simple method of measuring modulus in other crumpled systems. With this newly found understanding, crumples can now be reliably designed to serve similar roles as those traditionally filled by foams. The adoption of crumpled matter in material science will spur the development of a host of new, light-weight compact materials from any material (metal, polymer, carbon, ceramic,…) or system (electronics, photovoltaics, microfluidics,…) that can simply be made thin.

## Methods

**Polycarbonate**. PC was used as received from Scientific Polymer Products and was reported to have a molecular weight of 60,000 Daltons. Solutions were created by dissolving the polymer in chloroform (Fisher Scientific, Optima grade) to various weight percents up to 10%. Nile red, a fluorescent dye, was often added to solution in trace amounts. Films were created in several ways. Below ~2 microns in thickness, films were created by spin coating solution on freshly cleaved mica supports. Instabilities limit the thickness in this case. Larger thicknesses were created through drop casting polymer solution on freshly cleaved mica supports in a chloroform-saturated environment, which was allowed to slowly evaporate over several days. Drop casting was limited to thicknesses above ~2 μm due to dewetting instabilities, which occurred during the casting process.

After creation, films were annealed for ~1 h at a temperature of 453 K in order to remove any residual stress caused by the sample preparation techniques. Films were scored with a scalpel blade, then released on a Milli-Q water surface. Film thickness was measured with atomic force microscopy (<2 μm) or was measured with confocal microscopy (>2 μm). Each film was measured in several locations and an average thickness was used. Variation was typically 12%.

**Polydimethylsiloxane**. Elastomeric PDMS films were created with sylgard 184 (Dow Corning) mixed in a 10:1 ratio. Films were cast on glass slides, which were covered in a thin layer of poly(acrylic acid) through spin coating a 5% by weight water/poly(acrylic acid) solution. PDMS mixtures were degassed in a vacuum oven, then coated through drop casting or spin coating on a poly(acrylic acid)-coated glass slide. Films were then placed in the vacuum oven and annealed for 1 h at 353 K under vacuum. Films were cooled to room temperature before use.

Films were scored with a scalpel blade and released on a Milli-Q water surface. Films were removed from the water surface, dried, and immersed in a toluene Nile Red solution. After a short time (~10 min) films were removed from the toluene solution, excess solution was removed from the films surface, and the film was allowed to dry. Once dry, films were stored in the flat state for 24 h before use. Thicknesses were measured in the same manner as the PC films outlined above.

PDMS films are tacky when produced in the absence of ultraviolet exposure or dirt. To remove adhesion from the problem, we coat the PDMS films with a randomly oriented monolayer of microparticles. Polydisperse PS colloids of average diameter 7 μm were used, or more inexpensively, dry cornstarch particles were used. Films were immersed in particles, then excess particles were removed with a Kim-Wipe. Monolayers were directly observed with the confocal microscope, ensuring that no excess of free particles were present during the experiment; all particles adhered to the film. Films after a crumpling experiment did not have a noticeably different surface coverage of particles.

**Crumpling**. Thick films were crumpled by hand, and no attempt was made to create repeatable crumples. Gloves were worn to ensure no contaminants were added during the crumpling process. Typically two points on the film edges are pushed a random distance towards one another. Then one hand holds these first two points as the second hand repeats the process in another random direction. As these steps are repeated, the film slowly compacts into the crumpled state. Once the approximately spherical ball reaches a desired level of compaction, it is loaded between the parallel plates of the experimental apparatus.

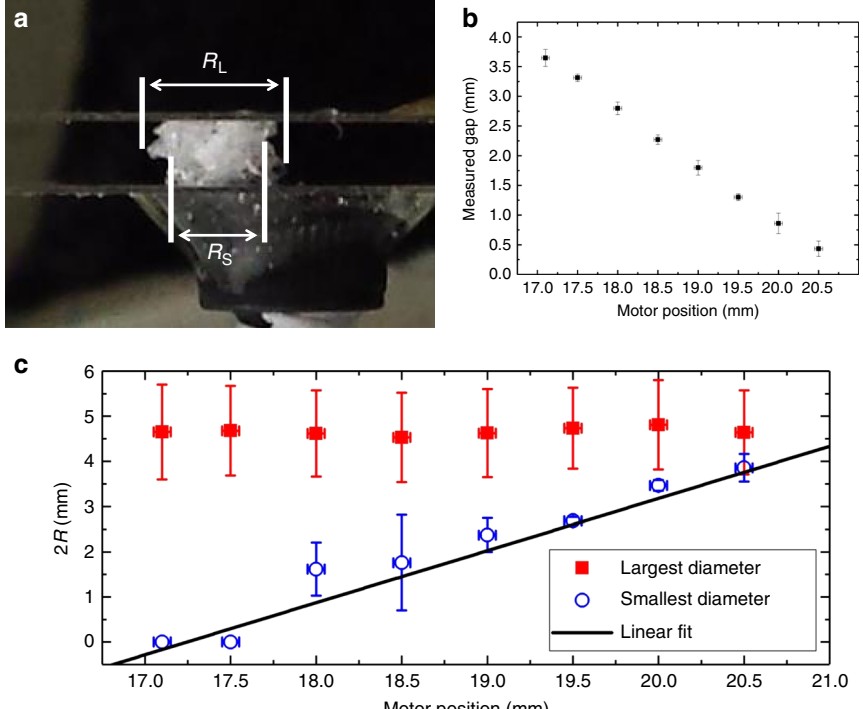

**Fig. 8** The horizontal extent of a crumple during compaction. **a** A typical image of the horizontal extent of a crumple. In this case, a polycarbonate film of diameter 4.7 mm is shown. $R_L$ and $R_S$ refer to the largest radius and smallest radius observed in the image. **b** Plate separation as a function of motor position showing a linear trend indicating that the apparatus is still stiff compared to the crumpled film. Error bars determined from the standard deviation of multiple measurements (Measured Gap) or motor step size (Motor Position). **c** Change in the largest (solid squares) and smallest diameter (open circles) as a function of compression. The larger diameter does not change during the experiment; however, the smaller diameter does grow in an approximately linear manner (a slope of 1.15 was found with a linear fit shown as the solid line). Error bars are the standard deviation of multiple measurements from two orthogonal images (Radius) or motor step size (Motor Position)

The thinnest samples required extra care. In this case, films are first floated off the substrate on which they were created, by immersing the substrate into a bath of pure (Milli-Q) water. Films are then removed from the water bath on a Kim-Wipe, in order to ensure the films do not collapse due to capillary effects. The film, supported by a Kim-Wipe, is then placed on a stack of dry wipes, which draws the remaining water from the supporting Kim-Wipe. Films are then delicately peeled from the Kim-Wipe with acetone-cleaned tweezers. Finally, the (now dry) film can be crumpled in the same manner as the thicker films, although fingers are now replaced with tweezers.

**Crumple radius**. The set-up used in this experiment has no bounding walls in the horizontal direction. This means that the definition of the lateral extent of the crumple (R) is ambiguous without further clarification. We adopted a definition based on the average lateral extent of the crumple as measured in images taken on two orthogonal axis through the side of the crumple, or via a two-dimensional projection of the crumple taken from the top with the confocal microscope. The side images were used preferentially as the smallest radius and largest radius in the images could easily be defined. Both smallest and largest radii were measured independently twice, along two orthogonal axis and averaged to a single value for R. Each image was calibrated independently ensuring no drift in scale. Typical results are shown in Fig. 8.

The largest radius did not change significantly during compaction for any of the materials studied. The smallest radius did often grow during compaction, indicating that a densification of the structural network occurs during crushing of at least some of the crumples studied. No correlation was noted between the rate of growth of the smallest radius and scaling exponent ($\alpha$ in the text), force amplitude ($F_0$ in the text) or time constant in force relaxation experiments.

## Data availability

The data that support the findings of this study are available from the corresponding author upon request.

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

## Acknowledgements

The authors gratefully acknowledge support from the AFOSR under the Young Investigator Program (FA9550–15–1–0168). T.E. thanks the NDSU Materials and Nanotechnology program for support. In addition, the authors thank Sean Gunderson for help analysing data.

## Author contributions

A.B.C. conceived of the work, conducted the experiments, analysed data and wrote the manuscript. T.T. and T.E. conducted experiments, analysed data and contributed to the writing process.

## Additional information

**Competing interests:** The authors declare no competing interests.

