## [Peer Review File · Nature Communications]

Reviewers' comments:

Reviewer #2 (Remarks to the Author):

This review is regarding the manuscript by Croll and co-workers entitled "Crumpled Matter". I read the paper with great enthusiasm: it is clear, well written, and on a topic that has broad impact and appeal. As such, I start with my conclusion that this work should be published in Nature Communications. The authors present a remarkably robust empirical model which provides excellent fits to the data. It is equally remarkable that the same model can be used to account for both the compression of a crumple and ridge. I have no major concerns and below only provide some comments and questions the authors may wish to elucidate.

p2: "For example, it remains unclear how much load, F , can be supported by a sheet of size L crumpled into a ball of radius R . " Without defining the geometry of the system (i.e. square sheet? Circular? How is the load applied?) I find defining the symbols a bit odd. They should be tied to the specific system.

p4: "Namely, polycarbonate (PC), a glassy polymer with a high modulus and polydimethylsiloxane (PDMS) an elastomer with a moderate modulus. " Would be nice to provide numbers here as it is not clear to me what constitutes a high and a low modulus.

p4: In general the manuscript is easy to read, however, the authors may wish to tighten up some of the writing, similar to the point above (moderate vs high without definition. Likewise "but a very different and much quicker retraction curve." — how is "very different" different from "different", "quicker retraction curve" than what? There is no mention of speed in figure 2, so I don't understand the meaning of a quick retraction curve.

How are the thin films crumpled into a ball? Especially the thinnest films?

p4: it is assumed that the crumpled sheet is a cylinder (line 72), but it is referred to as a crumpled ball in other spots. Perhaps the assumption here is that the cylinder has its symmetry axis along the direction of the applied force, in which case the ball is compressed into a "cylindrical" pancake structure. If I am right, then I think that the reader could be aided with a bit better explanation, especially since assuming that the spherical structure is treated as a cylinder is quite an important assumption.

p4, l73: Fig 2f is not sufficiently explained. The data is normalized using H_0 and ρ_0 , however, these have not been defined.

p6: Eqn 1 "is quantitative in fitting data.". I don't understand the meaning of this? What does it mean for the eqn to be quantitative in fitting the data? Is the data fit well by the eqn? I am confused because even a bad fit is quantitative.

p8: how are the geometric parameters obtained along the s direction in fig 4?

Though equations 1 and 2 are empirical in nature, it would be good to have some discussion as to the origin/basis given that the authors must have some intuition on this.

Reviewer #3 (Remarks to the Author):

Referee report concerning « Crumpled matter » by Croll et al.

This paper concerns mostly an experimental work aiming to establish scaling laws on highly crumpled matter. Elementary structures as bending, folding, D cone and ridges are presented and recent theoretical works or rather recent are reminded. It turns out that the experimental results of the authors, working on a large number of decades do not agree completely with previous theoretical models. A new scaling is proposed which seems to be better, resting on ridges only.

The topics seems to be fashionable since a similar study has been published before the submission by Croll et al.

“A state variable for crumpled thin sheets” has been submitted in November (9) 2018 in Communications Physics by Gottesman et al. Before, this paper was available on Archiv. It was also mentioned recently in NYTimes, so it is difficult now to ignore it. Due to this recent context, it is important to position the present work with this publication.

About the paper itself, the bibliography is really not adequate. I would like to mention that the work D-cone has been introduced for the first time (at least it seems to me) in “ Papier froisse, La Recherche “ (1995) and in Crumpled paper in 1997 by Ben Amar and Pomeau and it becomes since then a common name. However a reference to these authors must be done at the first page. [14] and [15] were not the first publications where this word appear and where D cone were explained as a fundamental building block.

For the paper itself, I have problem to understand what is $\$H\$$ and $\$X\$$ for the unique ridge. A more detailed description of these variables is necessary, especially due to the importance that the authors give to the ridge. Since the authors think that the important and relevant compressive unit is the ridge, how do they expect the ridges to enter in the full crumpling. In other words, what is the statistics of ridges in highly crumpled matter.

The conclusion of the paper is a little short. If place is missing, I do not think that Fig.1 which is rather huge is really necessary. In my opinion it is necessary to also give some theoretical interpretation of Eq.(1) or at least Eq.(2).

So I do not think that this paper can be published in its present form.

This review is regarding the manuscript by Croll and co-workers entitled “Crumpled Matter”. I read the paper with great enthusiasm: it is clear, well written, and on a topic that has broad impact and appeal. As such, I start with my conclusion that this work should be published in Nature Communications. The authors present a remarkably robust empirical model which provides excellent fits to the data. It is equally remarkable that that the same model can be used to account for both the compression of a crumple and ridge. I have no major concerns and below only provide some comments and questions the authors may wish to elucidate.

Thank you for the kind words and for sharing our excitement! We are thrilled you found our work to be of the high quality necessary for publication in Nature Communications. We are sure your efforts will result in an even stronger publication.

p2: “For example, it remains unclear how much load, F , can be supported by a sheet of size L crumpled into a ball of radius R . ” Without defining the geometry of the system (i.e. square sheet? Circular? How is the load applied?) I find defining the symbols a bit odd. They should be tied to the specific system.

This was an attempt to keep the word-count down but in hindsight this does seem a bit awkward in wording. In the current format we have a larger word count so we have revised this wording and clarified the definition of these variables.

p4: “Namely, polycarbonate (PC), a glassy polymer with a high modulus and polydimethylsiloxane (PDMS) an elastomer with a moderate modulus. ” Would be nice to provide numbers here as it is not clear to me what constitutes a high and a low modulus.

You are right again, this is needlessly vague. We have added the precise values for clarity.

p4: In general the manuscript is easy to read, however, the authors may wish to tighten up some of the writing, similar to the point above (moderate vs high without definition. Likewise “but a very different and much quicker retraction curve.” — how is “very different” different from “different”, “quicker retraction curve” than what? There is no mention of speed in figure 2, so I don't understand the meaning of a quick retraction curve.

Again we apologize for this terrible wording. We hope the revisions make the language much more clear. Thank you for taking the time to point this out for us.

How are the thin films crumpled into a ball? Especially the thinnest films?

A good question. We intentionally avoided creating any explicit crumpling protocol because we felt that it was important to have truly random crumples. Our thought was that if we repeated the exact same procedure, we would likely limit the different networks created and miss making important observations. We also each created our own crumples resulting in more variation. Working with the very thin films did require us to develop a few new techniques in order to ensure that films were not “artificially” collapsed by capillary effects. This should be described in more detail, as you and likely many other researchers might be interested in our sample preparation techniques. We have added the following description to the Methods section (which we can now add to the paper) to help clarify:

“Thick films were crumpled by hand, and no attempt was made to create 'repeatable' crumples. Gloves were worn to ensure no contaminants were added during the crumpling process. Typically two points on the film edges are pushed a random distance towards one-another. Then one hand holds these first two points as the second hand repeats the process in another random direction. As these steps are repeated, the film slowly compacts into the crumpled state. Once the approximately spherical "ball" reaches a desired level of compaction, it is loaded between the parallel plates of the experimental apparatus.

The thinnest samples required extra care. In this case films are first floated off the substrate on which they were created, by immersing the substrate into a bath of pure (Milli-Q) water. Films are then removed from the water bath on a Kim-Wipe, in order to ensure the films do not collapse due to capillary effects. The film, supported by a Kim-Wipe, is then placed on a stack of dry wipes which draws the remaining water from the supporting Kim-Wipe. Films are then delicately peeled from the Kim-Wipe with acetone cleaned tweezers. Finally, the (now dry) film can be crumpled in the same manner as the thicker films, although fingers are now replaced with tweezers.”

p4: it is assumed that the crumpled sheet is a cylinder (line 72), but it is referred to as a crumpled ball in other spots. Perhaps the assumption here is that the cylinder has its symmetry axis along the direction of the applied force, in which case the ball is compressed into a “cylindrical” pancake structure. If I am right, then I think that the reader could be aided with a bit better explanation, especially since assuming that the spherical structure is treated as a cylinder is quite an important assumption.

Yes, this important point is not clear in the previous draft. We have made some changes in order to explain this more thoroughly in the manuscript. It is important to note that the only place the cylindrical geometry is assumed is in analysis is in the calculation of the ‘effective’ modulus of the crumples (fig. 2f.). All other analysis does not make any assumption of the overall shape, although, as described in the supplementary file a maximum radius is measured for each crumple.

p4,173: Fig 2f is not sufficiently explained. The data is normalized using H_0 and ρ_0 , however, these have not been defined.

Unfortunately we edited these definitions out of the main text of the previous draft. We have added them back in the figure caption as they are definitely necessary for understanding the figure. This is a standard ‘light-weight materials’ plot, the two values are Young’s modulus of the sheet material (i.e. polycarbonate) and the density of the sheet material.

p6: Eqn 1 “is quantitative in fitting data.” I don’t understand the meaning of this? What does it mean for the eqn to be quantitative in fitting the data? Is the data fit well by the eqn? I am confused because even a bad fit is quantitative.

We were trying to use connective language to our earlier statements regarding the non-quantitative nature of the other scaling models. Scaling by design often ignores factors of order 1, but the previous models were off by several orders of magnitude. In these cases, ignoring the other flaws, the models only fit if non-physical values of modulus are used (the prediction is non-quantitative; it predicts the wrong quantity). Clearly this wording is not helpful. Maybe exact is a better word? We have made this change to the text. Our model is exact; we can recover the true material modulus.

p8: how are the geometric parameters obtained along the s direction in fig 4?

The work is carried out under a confocal microscope, which is used to locate the film’s 3D position in space. This allows the curvature at positions along s to be measured directly. The height of the film along s allow the second primary curvature to be directly measured. We have added a clarifying comment to the figure captions explaining the origin of these measurements.

Though equations 1 and 2 are empirical in nature, it would be good to have some discussion as to the origin/basis given that the authors must have some intuition on this.

We try to keep any opinion out of the publication, as our goal is to supply reliable data to the theorists working in the field and to give the engineering community a trustworthy structure-property relation. While we have some ideas as to the origin of the scaling, we do not yet have direct proof, so opted to avoid any embarrassing discussion were our opinions to be incorrect. We feel our work will spurn much more thorough discussion within the community than we could add here. The data shows the empirical model works, regardless of interpretation.

As both reviewers seem to desire a comment, we have added more discussion to the paper. As we lay out in the paper, we believe bending is playing a major role in the energy storage in the ridge. This is not to say stretching does not occur in the ridge; it is well established that there is stretching occurring in the ridge. It is simply the case that bending in the structure still overwhelms the stretching costs. The flaw of the original ridge models is simply that they underestimate the total bending in the sheet surrounding the ridge. The force scales as Eh^2 , rather than Eh (stretching) or Eh^3 (bending) – as bending is driven down to curvatures of order $1/h$ and no further (as suggested by Deboeuf). Again folds are not present, so the d-cone cores must be where this high energy bending occurs.

The bending cannot smoothly drop to zero along the ridge as it eventually interacts with the bending caused by the second d-cone. As the ridge is stressed, it is perturbed from its initial configuration of length X_0 , causing this ‘confinement’ to overlap. As the system is perturbed from the initial state a function $f(X_0/X)$ can be constructed to describe the response. In the PDMS the overlap is not a large cost, but enough to keep the structure from spontaneously decaying (e.g. a snap through event). In the PC ridges, the d-cones are unable to move freely and curvature does not change quickly. Energy is built up around the d-cone core as it slowly deforms, or as is more likely the case the ridge buckles releasing some of the stress around the d cone.

The PC data of our current setup is not quite of high enough resolution to see these small changes in the d-cone core. It is definitely an interesting question for future work.

Reviewer #3 (Remarks to the Author):

Referee report concerning « Crumpled matter » by Croll et al.

This paper concerns mostly an experimental work aiming to establish scaling laws on highly crumpled matter. Elementary structures as bending, folding, D cone and ridges are presented and recent theoretical works or rather recent are reminded. It turns out that the experimental results of the authors, working on a large number of decades do not agree completely with previous theoretical models. A new scaling is proposed which seems to be better, resting on ridges only.

Thank you for your efforts to review our work. We are sure your comments will help strengthen our manuscript.

The topics seems to be fashionable since a similar study has been published before the submission by Croll et al.

“A state variable for crumpled thin sheets” has been submitted in November (9) 2018 in Communications Physics by Gottesman et al. Before, this paper was available on Archiv. It was also mentioned recently in NYTimes, so it is difficult now to ignore it. Due to this recent context, it is important to position the present work with this publication.

We did see this paper come out, but we had submitted our work before it was published. We didn't notice the arXiv version beforehand, but this paper does not mention our even earlier arXiv post outlining some different aspects of an early approach to our model (<https://arxiv.org/abs/1801.01166>). Our earlier work is not published (or submitted at this point as we need to rewrite it). In short, I think we can all agree that searches don't always find arXiv works right away.

Having read this work in the intervening time we can certainly comment on it. The authors of the new paper repeat the experiment done many years ago by Blair and Kudrolli (PRL, 166107, 2005) in which a crumpled sheet is uncrumpled and its topography scanned. In both works, the remaining deformation is reduced to 'folds' and the length of each fold is measured. The new paper focuses on the total length of folds, rather than the distribution of lengths as originally measured by Blair. Interestingly, the new work does not compare their length distributions to the original work (although they clearly have these details), so it is hard to tell how to compare mylar of the recent work to the paper used by Blair. In fact, they barely discuss the Blair results.

The work involves a completely uncontrolled system - store bought plastic (mylar) of one thickness which may have any number of unknown material properties. For example consider this short list of obvious issues: residual stress or curvature from processing and shipping is not known, an unknown aging state (as it is a glassy polymer), unknown content of residual solvent, unknown defect content (e.g. dirt), unknown uv-degradation, or thermal degradation, and so on. They make no effort to understand how failure occurs in the material, though their main "measurement" is a measurement of material failure (residual curvature after crumpling). This superficial approach deeply concerns us.

While the plastic damage is an interesting part of a crumple, it is not truly representative of the actual structure that is present in a crumple. For example, the elastic sheets we use would give a null result by their measurement regardless of the crumpling that took place beforehand. Studying plastic damage alone is an indirect 'easy' experiment that only shows a 'ghost' of the true crumple structure. Interestingly this fact was pointed out in the excellent reference you recommend below. From Ben Amar and Pomeau "... for what reason do folds become permanent and what other deformations, a priori, are equally important ..." apologies if my translation is imperfect. The point is that folds are not necessarily the only feature relevant to the final mechanics of the object. We prove this fact by showing elastic materials behave almost identically under our experimental conditions.

All that said, I don't doubt that Gottesman et al. found an interesting relationship for the total damage done during crumpling of Mylar. The results, for the particular thickness of Mylar they bought, seem robust. It is a nice result, but certainly not universal as, by design, it cannot apply to an elastic sheet. And ultimately not practical – the total length of plasticity is not directly related to the forces applied to the structure. This means that the result, in its current state, is not useful to the engineering community as it does not help predict system behaviour. As you suggest, we will add a citation but will not discuss the work in detail due to space considerations and because it does not directly relate to our publication which focuses on force measurement.

To directly answer the editor's question above, our work allows someone to use an average (statistically determined) power law exponent to predict the exact force that a crumpled ball of any material will hold. This is an important, practical, result that will enable a crumple to be

treated as any other engineered structure. It will make crumples useful. The communications physic paper does not allow any practical prediction of useful macroscopic properties; it simply examines the location and extent of plastic deformation. It is an interesting statistical result, which gives some insight to the network of plastic failure in crumpled plastic sheet, but does not make any connection to how force is transported through the network. This is very different from the question we address.

Additionally, our work does not pre-suppose that plastic deformation is the only cause of strength in the structure (as we prove using a non-plastic material). We work very hard to control the materials from start to finish, ensuring that any contamination that exists in the material is minimal and understood. We ensure our glassy films are annealed prior to use in order for each material to have the same thermodynamic history, be free of residual stresses (and curvatures) and to ensure the entanglement network is distributed uniformly. We have painstakingly measured all relevant material properties of the systems we use with independent methods (see, elder et al. macromolecules DOI: 10.1021/acs.macromol.8b02002) to ensure we understand the system we are working with. We use many orders of magnitude in thickness to make sure we don't have a result that only applies in some narrow limit. Our results show consistency over 50 orders of magnitude.

About the paper itself, the bibliography is really not adequate. I would like to mention that the work D-cone has been introduced for the first time (at least it seems to me) in “ Papier froisse, La Recherche “ (1995) and in Crumpled paper in 1997 by Ben Amar and Pomeau and it becomes since then a common name. However a reference to these authors must be done at the first page. [14] and [15] were not the first publications where this word appear and where D cone were explained as a fundamental building block.

We apologize for the oversight; we were aware of the 1997 paper but inadvertently left it out of the bibliography. The earlier paper we did not know of, so are greatly indebted to the reviewer for pointing out this work. This is a very nice, general audience, paper and we quite enjoyed reading it (although I am sure my rough translation is imperfect). We have included citations to both in our manuscript, as well as a few others regarding the statistical distributions measured in similar problems.

For the paper itself, I have problem to understand what is $\$H\$$ and $\$X\$$ for the unique ridge. A more detailed description of these variables is necessary, especially due to the importance that the authors give to the ridge. Since the authors think that the important and relevant compressive unit is the ridge, how do they expect the ridges to enter in the full crumpling. In other words, what is the statistics of ridges in highly crumpled matter.

We have added clarification of our variables in the text. X refers to the original length of the ridge (eg. The d -cone to d -cone distance at the start of the experiment), H refers to the particular gap of the cell at some moment in the experiment. X simply replaces R_0 for the full crumple, so we have relabeled X as X_0 to help clarify. To connect the number and position of ridges to the overall crumple is a very interesting challenge, and we are sure our data will help theorists to develop and test more comprehensive models. Our goal is to supply reliable data to the community, we don't wish to sully the work with claims grander than this.

As this is a review, we can offer some untested hypothesis to wet your appetite. We imagine that the crumple is built of many 'springs' in series or parallel. The question is then of the network structure in which the springs (ridges) are added up. For example, imagine a set of

n identical ridges bears the load from one side of the crumpled ball to the other (a force chain of sorts). If each ridge were identical, the force in the ridge would be written $F = Eh^2 X^a / H^a$, where *a* is the particular exponent measured and *H* is the level of compression on the particular ridge (eqn. 2 in our paper). The crumple size $R_0 = X + X + \dots + X = nX$ where *n* is the number of ridges in the chain of ridges, and $H_c = H + H + \dots + H = nH$ is the total gap in which the crumple is confined. The effective force (the whole crumple) we would write $F_c = Eh^2 (R_0^a / H_c^a)$ or substituting for R_0 and H_c , $F_c = Eh^2 (nX/nH)^a = F$ as it must. The ridge itself is similar; it is a collection of many different bent segments of film operating at different lengthscales in concert.

Obviously, this example is very simplified and much more may occur in real structures. For example, in the glassy PC films, the ridges do not observably deform before they buckle. This would imply some parts of the 'network' of load bearing pieces are effectively solid. In this case exponents get much higher, as you can easily prove to yourself in considering several Hookean springs (for simplicity) in a network, and then replacing several with solid elements.

The question you ask, that of the statistics, is a very interesting question. Many excellent works exist which focus on measuring ridge statistics, but are typically aimed at plastic materials (e.g. counting creases as noted above). We have only examined the statistics of the measured exponents directly in our work, and this appears in the supplemental information. Interestingly, the elastic materials seem to have exponents distributed with a Gaussian shape, whereas the plastic materials seem to show log-normal type shapes (which would be similar to the ridge counting statistics of Blair and others). We believe that this means the network of ridges is directly related to the observed exponent, though proof of this claim is beyond the scope of our current work. We are sure many researchers will be interested in fitting statistical models to our data and are excited to see the results. We are certain our work will have a greater impact because of the discussion that it will create.

The conclusion of the paper is a little short. If place is missing, I do not think that Fig.1 which is rather huge is really necessary. In my opinion it is necessary to also give some theoretical interpretation of Eq.(1) or at least Eq.(2).

Unfortunately, it was a word limit we were facing rather than a physical space limitation (before the manuscript was transferred to Nature Communications). We feel figure 1 is important for a general audience; it is our experience that a picture of these structures is the only clear way to describe them. Certainly it could be made smaller during publication without altering its purpose. We have added additional discussion now that the space limitations are less restrictive; we hope you find this satisfactory. Ultimately, this paper will create new discussion and lead to much more detailed and lengthy publications down the road.

So I do not think that this paper can be published in its present form.

We hope we have convinced you that the work is now of sufficient quality for publication in Nature Communications. Regardless, we would be happy to continue this discussion with you if there are additional questions raised. Thank you again for your efforts on our behalf.

REVIEWERS' COMMENTS:

Reviewer #2 (Remarks to the Author):

The authors have responded to all concerns with care. I recommend publication of this work in Nat. Comm.

Suggestion: "is exact in fitting data."  "provides an excellent fit to the data"?

Regarding the extensive discussion, the authors state "As both reviewers seem to desire a comment, we have added more discussion to the paper." I think this is great and warranted. I would advice adding a qualifier to the discussions i.e. making it more clear that you provide a tentative explanation but more work is required.